# The Interaction Mechanism of Fiscal Pressure, Local Government Behavioral Preferences and Environmental Governance Efficiency: Evidence from the Yangtze River Delta Region of China

**DOI:** 10.3390/ijerph192416618

**Published:** 2022-12-10

**Authors:** Tinghui Wang, Qi Fu, Yue Wang, Mengfan Gao, Jinhua Chen

**Affiliations:** 1School of Politics and Public Administration, Soochow University, Suzhou 215123, China; 2The Institute of Regional Governance, Soochow University, Suzhou 215123, China; 3Research Institute of Metropolitan Development of China, Soochow University, Suzhou 215123, China; 4School of Urban and Rural Construction, Shanxi Agricultural University, Taigu, Jinzhong 030801, China

**Keywords:** fiscal pressure, local government behavioral preferences, environmental governance efficiency, interaction mechanism, Yangtze River Delta region

## Abstract

In environmental governance, local governments are the main actors, and their behavioral preferences between economic growth competition (EGC) and environmental regulation (ER) affect the inputs and outputs of environmental governance. Most studies discuss the relationship between government behaviors and the environment from the fiscal decentralization perspective, with few studies from the fiscal pressure (FP) perspective. Importantly, the bidirectional interaction mechanism is easily ignored. This study measured local government FP, EGC, ER, and environmental governance efficiency (EGE) in China’s Yangtze River Delta (YRD) region from 2000 to 2020. Moran’s I index was used to identify the change characteristics of local government behavioral preferences. The interaction mechanism was analyzed by a panel vector autoregression (PVAR) model. The results show that (1) from 2000 to 2020, FP was generally strengthened. EGE generally showed fluctuating and rising change characteristics, with more obvious fluctuating and rising characteristics before 2012 and after 2012, respectively. Local governments shifted from a strong alternative preference to a weak synergistic preference. (2) FP had a self-reinforcing effect. EGC and ER had a self-weakening effect. EGE had not only a self-weakening effect but also a weak self-dependence. (3) There is a double negative interaction mechanism between FP and local government behavioral preferences. FP made local governments prefer to reduce EGC and relax ER, but in fact, EGC and ER were conducive to alleviating FP. (4) There is a negative transitive influence mechanism between FP, local government behavioral preferences and EGE. The negative effect of FP on EGE can be transmitted by reducing EGC and ER. This paper provides a scientific basis for improving EGE in the YRD region and understanding the behavioral logic of local governments’ environmental governance and a reference for other rapidly industrializing and urbanizing regions.

## 1. Introduction

The industrialization of modern countries has brought economic growth, but it has also created environmental problems. Environmental pollution is an inevitable problem faced by any country or region in the process of industrialization [1,2]. As a public good, the environment is nonexclusive, noncompetitive, and external. The protection and governance are mainly provided by governments. However, due to the differences in system backgrounds, economic structures and public policies, governments in different countries and regions face different practical dilemmas in environmental governance [3]. Thus, there are different behaviors and preferences regarding environmental governance, which affects its effectiveness [4,5,6].

In China, the reform of the fiscal and taxation system starting in 1994 has reduced local governments’ share of fiscal revenue. Meanwhile, the decentralization of public affairs has led local governments to undertake numerous public functions. Due to the centralization of finance and the decentralization of administration, local governments must undertake more public affairs with limited financial resources, resulting in the dilemma of fiscal pressure (FP) [7]. In addition, regional differences in resources, locations, and policies are difficult to resolve. The conflict between the limited financial resources of local governments and fiscal expenditures has been further sharpened [8,9]. In China, environmental problems are mainly caused by pollution emissions from industrial enterprises [10]. However, since industry supports local economic growth and GDP is an important indicator in the assessment system for local governments and officials in China, there is a competitive relationship between local governments [11], resulting in the economic growth competition (EGC) of local governments. Moreover, the tax payments of industrial enterprises are an important source of local fiscal revenue, and there is an interest alliance relationship between local governments and enterprises [12]. Therefore, local governments that lack restrictions and constraints tend to relax environmental regulation (ER), that is, relax the entry threshold and regulatory standards for high-pollution enterprises, thereby seeking economic growth and increased fiscal revenues. This preference of local governments is an important reason why environmental pollution is endless and difficult to completely eradicate [13,14].

Efficiency is the core value of public administration. For a long time after the reform and opening up, the Chinese government paid more attention to effectiveness rather than efficiency in governance. This choice had realistic rationality in the country’s development stage. However, as mega public organizations, governments need to regain their focus on the value of efficiency in the modernization process. The most intuitive and essential expression of efficiency is the effective outputs of inputs. The value of efficiency also needs to be emphasized in environmental governance. Environmental governance efficiency (EGE) involves measuring the effective use of inputs by the output effects of environmental governance [15,16]. It is easy to ignore the manpower and financial investment of local governments by focusing only on the changes in environmental pollution indicators [16]. Measuring the EGE from the perspective of inputs and outputs is conducive to reflecting the effectiveness of environmental governance in a more scientific and comprehensive manner [17].

The impact of FP on EGE is manifested in the fact that FP will affect the behavioral preferences of local governments between EGC and ER, which in turn, will affect the inputs and outputs of environmental governance. For example, some studies found that local governments with fiscal autonomy are more concerned with economic growth and the expansion of financial resources [18] so that they can reduce the pressure of competition and stand out in political promotions [19,20]. This phenomenon has also led to a lack of sufficient financial investment guarantees and willingness of local governments to engage in environmental governance [21,22]. At the same time, local governments tend to lower the entry barriers for companies to introduce overcapacity and high-pollution enterprises [23,24]. Existing taxpayers in enterprises can also avoid environmental supervision and control [12,25]. These factors lead to an increase in the effects of environmental pollution and prevent the available resources from having a real impact on governance. However, the studies cited above are mostly based on the fiscal decentralization perspective rather than the FP perspective. Additionally, fewer of them place the elements above in the same framework to confirm the influence of local government behavioral preferences on EGE. Importantly, these effects may also exist in the reverse dimension, but this interaction mechanism is easily overlooked.

Since the 18th National Congress, China’s central government has vigorously promoted the national strategy of ecological civilization construction, making environmental protection and governance a compulsory public affair of local governments. Along with the implementation of the one-vote veto system and central environmental protection supervision, environmental protection, like economic growth, has gradually become an important constraint for administrative assessment and political promotion. In the face of major changes in systems and policies, will local governments strengthen the supervision and control of polluting enterprises to avoid the accountability of and punishment by the central government? What are the characteristics of the local government behavioral preferences between EGC and ER? How will FP and EGE change? These are issues that have strong research implications but have received less attention and discussion in the past.

The Yangtze River Delta (YRD) region is one of the most active and open regions in China in terms of economic development. However, rapid urbanization and population agglomeration bring a considerable amount of public affairs; thus, local governments may also face FP. Along with rapid industrialization, the inflow of capital and the concentration of industries have brought environmental pollution that is not optimistic. This situation is more likely to cause changes in the behavioral preferences of local governments between EGC and ER. Therefore, with rapid industrialization and urbanization, the YRD region is more representative and realistic.

In summary, this study uses China’s YRD region as an example to conduct an empirical analysis to compensate for the limitations of previous studies. The study covers the following three main areas: (1) it analyzes the change trends of FP and EGE in the YRD region from 2000 to 2020; (2) it identifies the change characteristics of local government behavioral preferences between EGC and ER using Moran’s I index; and (3) it analyzes the impact of and response among FP, EGC, ER, and EGE based on a panel vector autoregressive (PVAR) model. Based on these components, this study discusses the change characteristics and interaction mechanism and proposes policy recommendations. This paper attempts to open the black box of government behavior in environmental governance and provides a scientific basis for improving EGE in the YRD region, harmoniously developing the economy and environment, and promoting ecological civilization construction. It also provides a reference for other rapidly industrializing and urbanizing regions.

## 2. Literature Review

Regarding the research perspective, previous studies have discussed the relationship between fiscal decentralization and environmental pollution from the decentralization perspective [19,20,21,22,23,24]. Fiscal decentralization refers to giving certain taxation powers and expenditure responsibilities to local governments, allowing them to determine the scale and structure of budget expenditures on their own. However, from a practical point of view, the 1994 tax-sharing system reform in China was essentially a centralized reform driven by the central government. The purpose was to keep central finance strong in central-local relations [26]. Through the reform of the “separation of finance and taxation”, local tax revenue is concentrated in the central government, and the central government then returns a certain proportion of local tax revenue through transfer payments. Not only the tax-sharing system reform in 1994 but also the subsequent income tax-sharing reform and replacement of business tax with value-added tax significantly reduced local government fiscal revenue [27], that is, disposable tax revenue. Therefore, China’s fiscal and taxation system is characterized by fiscal decentralization and tax centralization or expenditure decentralization and revenue centralization [28]. That is, local tax revenue is controlled and distributed by the central government, while the creation of tax revenue and fiscal expenditure is the responsibility of local governments [29]. Therefore, the reformed Chinese fiscal system is different from traditional fiscal federalism and is even characterized by centralization. As a result, it may be inaccurate to discuss China’s environmental pollution from the fiscal decentralization perspective alone.

Scientifically measuring fiscal decentralization is also a challenging task. To reflect the degree of fiscal decentralization, most of the empirical literature uses the proportion of expenditure and revenue of different levels of government [20,21,22,23,24,30,31,32]. However, the applicability of this measure is very limited because it ignores the complexities of central and local fiscal relations in fiscal decentralization, and is likely to cause serious distortions in the empirical findings [33]. Fiscal decentralization is determined by institutional design, policy instruments, power allocation and the organizational structure [34,35]. Thus, there is less variation in time and space within the same institutional and policy context. In addition, due to the existence of tax rebates and transfer payments, the degree of fiscal domination and freedom cannot be simply measured by the proportion of fiscal expenditure and revenue. This measurement essentially reflects the fiscal revenue and expenditure faced by local government in a short period of time, which makes it difficult to accurately reflect the degree of fiscal decentralization. We believe that the mismatch between finance power and administrative power has led to FP. The impact of fiscal decentralization on the environment that most empirical studies have focused on is essentially the impact of FP on the behavioral preferences of local governments. In summary, research and discussion based on the fiscal decentralization perspective have certain limitations in terms of both reality and methodology. Discussing the relationship between government behavior and the environment from the FP perspective may have more applicability and practical significance for China.

Some scholars have also paid attention to the importance of studying environmental issues from the FP perspective, but there are still few relevant studies. For example, Bai et al. argued that Chinese local governments have little or no formal taxation power. It is more appropriate to explain environmental pollution from the perspective of fiscal concentration and FP. Additionally, they analyzed the negative impact of local government tax competition on environmental quality using data from 30 provinces in China [27]. Kou et al. held a similar view and examined the local government regulation of SO2 under dual environmental pressure and FP using data from 30 provinces in China [36]. In addition, Zhang et al. confirmed a positive relationship between the scale of local government debt and environmental pollution based on Chinese provincial panel data [37]. Chang et al. demonstrated that local government FP can increase air pollution using panel data from resource-dependent cities in China [29]. Kong et al. found that the financial status of local governments in China is an important driver of environmental pollution control [38]. These studies provide a good foundation for this study. They focused on explaining the impact of a certain behavior of local governments on the environment under FP. Unfortunately, they did not explore the interaction between local government behavioral preferences and EGE.

From the research content perspective, most researchers believe that fiscal decentralization will promote economic growth and lead to environmental pollution, because fiscal decentralization gives local governments more freedom. Inspired by political promotion and administrative assessment, local governments have the motivation to grow the economy and the enthusiasm for maximizing tax revenue under the assumption of “ economic man” [18,19,20,21,22], that is, the theory of the political promotion tournament [11]. At the same time, the competition for the current capital represented by foreign direct investment (FDI) has led to a “race to the bottom” in the ER of local governments, which in turn has led to the aggravation of environmental pollution [23,24,25], that is, the pollution haven hypothesis [39]. These theories have certain realistic rationality in certain development stages of China. Under the theoretical assumption, most studies analyze EGC and ER as two opposing factors to explain the impact of fiscal decentralization on the environment. However, they are rarely included within the same framework to discuss the changes in and relationships between the two. In authoritarian China, the top-down decision-making mechanism and the resulting changes in institutions, structures, and policies have a great impact on the behavior of local governments [40]. In addition, the environmental kuznets curve shows that the relationship between the economy and the environment is not constant and irreconcilable [41]. Therefore, under the background that the central government is vigorously promoting ecological civilization construction and high-quality development, the behavioral preferences of local governments between EGC and ER may have changed. We argue that the same attention needs to be paid to the change characteristics and interaction of EGC and ER over time and space.

Moreover, our literature review reveals that studies have mostly discussed the relationship between finance, economics, regulation, and the environment based on unidirectional dimensions or a transitive relationship [19,20,21,22,23,24]. Less attention has been paid to the bidirectional and interactive relationship between these variables, affecting the perception of regularity. The behavior of local governments in environmental governance is complex and dynamic [25]. Probing the interaction mechanism among FP, EGC, ER and EGE is more conducive to opening the black box of local government behavior in environmental governance and understanding its logic.

From the perspective of research methods and scales, the time series data or panel regression models that have been widely used in research tend to ignore the endogeneity, complex dynamic relationships and lag effects of variables. A PVAR model allows all variables to be endogenous and can reflect the dynamic relationship of multiple variables. The introduction of variable lag terms can also effectively solve the endogeneity problem. In addition, the scale of related research is more concentrated at the provincial level. However, environmental governance is mainly the responsibility of local governments under the territorial management system. Additionally, because the upward obstruction and asymmetry of information, the behavioral preferences of local governments are more likely to change. As a result, we narrow the scale and choose the prefecture-level scale for research.

## 3. Materials and Methods

### 3.1. Study Area

The YRD region is located on the eastern coast of China and the lower reaches of the Yangtze River (Figure 1). It is an alluvial plain formed before the Yangtze River enters the sea. The YRD region consists of three provinces and a city with provincial status: Anhui, Jiangsu, Zhejiang, and Shanghai. There are 16 prefecture-level cities in Anhui Province, 13 prefecture-level cities in Jiangsu Province, and 11 prefecture-level cities in Zhejiang Province. In addition to the municipality of Shanghai, there are 41 cities (Table 1).

The YRD region is a rapidly urbanizing and industrializing region in China. It covers an area of 358,000 square kilometers, accounting for approximately 3.73% of China’s area. In 2020, the resident population reached 235 million, an increase of 109 million from 2000, accounting for approximately 16.64% of China’s population. The average urbanization rate reached 75.01%, higher than the Chinese average of 11.12%. The YRD region is also one of the most active and open regions in China. In 2020, it achieved a total regional GDP of 24.47 trillion yuan, an increase of 9.8% over 2000, accounting for 24.09% of China’s GDP. The secondary industry in each province accounts for more than 40% of GDP. In 2020, FDI in the YRD region reached US$ 3315.1 billion. In recent years, the central government has taken the lead in piloting ecologically friendly development in the YRD region. In this context, it is more practical to determine the changes and the interaction mechanism of FP, EGC, ER, and EGE.

### 3.2. Data Sources

The 2000–2020 panel data covering 41 cities in the YRD region are mainly from the “Anhui Statistical Yearbook”, “Jiangsu Statistical Yearbook”, “Zhejiang Statistical Yearbook”, and “Shanghai Statistical Yearbook”. They are supplemented by the “China Urban Statistical Yearbook”, “China Environmental Statistical Yearbook”, and the statistical yearbooks of various cities. Taking into account the availability, accuracy, and uniformity of the data, the time period of 2000 to 2020 was selected. Since this study focuses on the identification and interpretation of the interaction mechanism between variables, the 2000–2020 panel data can ensure that the data have a certain scale and that the empirical test of core issues has a longer period of time. It is also conducive to reflecting changes in institutions and policies.

### 3.3. Variables

(1)Fiscal pressure (FP)

As mentioned above, the measurement of fiscal decentralization in related studies essentially reflects the FP of local governments. There are three main indicators used to measure fiscal decentralization: revenue, expenditure, and autonomy. However, the first two have the problem of denominator identity; thus, they cannot reflect the cross-sectional information of the measurement results [42,43]. Therefore, the ratio of local fiscal expenditure to local fiscal revenue was used to reflect the intensity of FP. This ratio can not only capture intertemporal changes in FP but also help to reflect regional differences [17]. The measurement is as follows:FP = Fiscal Expenditure/Fiscal Revenue

(2)Economic growth competition (EGC)

The EGC of local governments was measured based on the theory of the political promotion tournament. Since the assessment and promotion of prefecture-level governments are controlled by provincial governments, they are a kind of relative performance evaluation that determines the characteristics of tournaments between local governments. Local governments need to surpass similar cities with similar attributes (level, GDP, etc.) to stand out, which in turn creates competition for economic growth. Therefore, local governments will not only pay attention to their own economic growth but also pay attention to other similar cities in the same province to seek higher economic growth to stand out in the fierce tournament. The following formula was used to measure the intensity of EGC:$$EGC_{i}=\sum^{\;}\frac{g_{j}}{\left|G_{i}-G_{j}\right|g_{i}},i\neq j$$
where i and j are different cities in the same province; $EGC_{i}$ is the economic growth competition intensity of city i in year t; $g_{j}$ is the GDP growth rate of city j in year t; $g_{i}$ is the GDP growth rate of city i in year t; and $G_{i}$ and $G_{j}$ are the per capita GDP of city i and city j in year t, respectively. This calculation represents the process rather than the result of the political promotion tournament. The larger the value is, the more disadvantaged the local government is in the tournament, and the greater the pressure to catch up.

(3)Environmental regulation (ER)

ER can be divided into two types, command-and-control and market-incentive regulation, but the goal is to regulate various behaviors that pollute the environment. In this paper, ER is defined as the various methods that local governments use to control the environmental pollution of industrial enterprises, and it is characterized by the results of the impact on the amount of emissions of enterprises. Considering the availability of data and referring to the ideas of previous studies [44,45], the intensity of ER was measured by two indicators: three industrial waste emissions (industrial wastewater emissions, industrial gas emissions, and industrial solid waste emissions) and the gross industrial output value. The emission of pollutants per unit of gross industrial output was used to measure the intensity of ER. The formula is as follows:$$ER_{it}=\frac{1}{n}\overset{n}{\underset{j=1}{\sum }}(p_{ij}/\frac{1}{m}\overset{m}{\underset{i=1}{\sum }}p_{ij})$$
where $ER_{it}$ is the environmental regulation intensity of city i in year t; n is the number of pollutants; m is the number of cities; and $p_{ij}$ is the ratio of the gross industrial output value in city i to the emission of pollutant j.

(4)Environmental governance efficiency (EGE)

EGE was measured by the input and output of environmental governance. The Super-SBM Model with undesirable outputs was used to measure the EGE. This model avoids the deviation caused by the failure of the traditional CCR model and the BCC model to take slack into account based on the radial and angle, effectively solving the problem stemming from the fact that the traditional DEA model cannot evaluate undesirable outputs. At the same time, it solves the problem created by the inability to compare and evaluate when the results measured by the SBM model are the same as full efficiency [46,47].

Referring to the viewpoints and ideas of similar studies [15,17,48] and considering the availability of data, two input indicators were set, manpower and finance, that is, government environmental protection practitioners and financial investment, respectively. The desirable outputs are the comprehensive utilization rate of industrial solid waste, the amount of industrial soot removed, the rate of sewage treatment and the gross industrial output value. The undesirable outputs are the amount of industrial solid waste generated, industrial waste gas discharged and industrial wastewater discharged. MATLAB R2020b was used to write the code to measure EGE. Descriptive statistics of the variables are shown in Table 2.

### 3.4. Methods

#### 3.4.1. Moran’s I Index

Moran’s I index is helpful in reflecting the degree of spatial correlation of elements [48,49]. It was used to identify the change characteristics of local government behavioral preferences between EGC and ER. Moran’s I index is as follows:$$I_{lm}^{p}=z_{l}^{p}\cdot \overset{n}{\underset{q=1}{\sum }}W_{pq}\cdot z_{m}^{q}$$
where $z_{l}^{p}=\frac{X_{l}^{p}-\bar{X_{l}}}{e_{l}}$; $z_{m}^{q}=\frac{X_{m}^{q}-\bar{X_{m}}}{e_{m}}$; $X_{l}^{p}$ is the attribute l of spatial unit p; $X_{m}^{q}$ is the attribute m of spatial unit q; $\bar{X_{l}}$ and $\bar{X_{m}}$ are the mean values of attribute l and attribute m, respectively; $e_{l}$ and $e_{m}$ are the variances in attribute l and attribute m, respectively. Moran’s I index covers [−1, 1]. When Moran’s I index is greater than 0, it means that there is a positive spatial correlation between the two variables. When it is less than 0, it means that there is a negative spatial correlation between the two variables. The space is random when Moran’s I index is 0. The larger the absolute value is, the stronger the spatial correlation.

#### 3.4.2. PVAR Model

The panel vector autoregression (PVAR) model was proposed on the basis of the vector autoregressive model (VAR). It combines the advantages of panel data analysis and the VAR model, and it can control the unobservable period and cross-sectional invariant effects and is conducive to analyzing the bidirectional dynamic relationship between variables. The advantage is that the empirical model is not based on economic theory and all variables in the system can be used as endogenous variables, thus avoiding problems such as model setting errors and endogeneity [50,51].

The model not only combines the advantages of time series and dynamic analysis but also can control for unobservable temporal and individual heterogeneity. It provides a more precise portrayal of the mechanism of the impact of independent variables on dependent variables under unidirectional and structural shocks. Considering the complex logical relationship among variables, the PVAR model was established to accurately reflect the interaction mechanism. The model was set as follows:$$Z_{it}=\Gamma_{0}+\overset{p}{\underset{j=1}{\sum }}\Gamma_{j}Z_{it-j}+f_{i}+d_{t}+\varepsilon_{it}$$
where $Z_{it}$ denotes the vector of endogenous variables for city i in year t, in the order FP, EGC, ER and EGE. j is the lagged order of the variable; $\Gamma_{0}$ is the intercept term; $\Gamma_{j}$ is the regression coefficient matrix; $f_{i}$ and $d_{t}$ denote individual fixed effects and time fixed effects, respectively; and $\varepsilon_{it}$ is the random disturbance term. The operation of the model was based on EViews 10 and Stata 16. The individual variables were logarithmized to avoid the problem of heteroskedasticity in the model estimation results [52,53].

## 4. Results

### 4.1. Trends and Characteristics of Changes

#### 4.1.1. Trends in FP and EGC

Between 2000 and 2020, most local governments in the YRD region were under FP (FP > 1 in individual years and FP > 6 in overall years) (Figure 2). Additionally, there was a general increasing trend in FP. The cities of Suzhou and Wuxi in Jiangsu Province and Hangzhou in Zhejiang Province were under relatively little financial pressure (FP ≤ 6 in the overall years). Local governments in Anhui Province faced a greater and more pronounced increase in FP compared to Jiangsu Province, Zhejiang Province, and Shanghai. The FP of local governments in Jiangsu and Zhejiang Provinces was relatively balanced. Local governments with less FP were concentrated in southern Jiangsu and northern Zhejiang and were closer to the level of Shanghai.

The EGE in the YRD region was assigned based on the mean value, and the change folding graph was plotted (Figure 3). As shown in Figure 3, the EGE of the YRD region showed a fluctuating and rising trend from 2000 to 2020, with more obvious fluctuating characteristics before 2012 and more obvious rising characteristics after 2012. The rankings of the mean value of EGE from 2000 to 2020 are as follows: Shanghai > YRD region > Anhui > Zhejiang > Jiangsu.

Among them, the EGE of Shanghai was at a relatively high level during the 2000–2020 period. The EGE of Anhui showed a fluctuating trend during the 2000–2012 period and an increasing trend during the 2012–2020 period. The EGE of Jiangsu generally showed an upward trend from 2000 to 2010. There was a plateau period from 2010 to 2016, followed by an upward trend. The EGE of Zhejiang showed a fluctuating trend during the 2000–2020 period.

#### 4.1.2. Characteristics of Local Government Behavioral Preferences

The GeoDa spatial analysis tool was utilized to calculate the Moran’s I indices between EGC and ER in 2000, 2004, 2008, 2012, 2016, and 2020 (Table 3). At the 95% confidence level, the six-period *p* values were all less than 0.05, passing the significance test.

The Moran’s I indices show that local government behavioral preferences between EGC and ER had a spatial agglomeration in the YRD region between 2000 and 2020. EGC and ER showed a negative spatial correlation and an increasing trend of change in the first three periods, indicating that local governments showed a relatively strong alternative preference. In the last three periods, a positive spatial correlation was shown in 2012 and 2020, and a negative spatial correlation was shown in 2016. However, all Moran’s I indices were relatively small. This result indicates that local governments were characterized by a weak synergistic preference overall. However, there was a fluctuation in 2016.

To further reflect the changes in the behavioral preferences of local governments, EGC and ER were compared with the regional average to distinguish between high and low with the reference to the ideas of previous studies [54,55]. The spatial and temporal distribution map of local government behavioral preferences was drawn using ArcGIS 10.2 (Figure 4). HH stands for high EGC and high ER; HL stands for high EGC and low ER; LH stands for low EGC and high ER; and LL stands for low EGC and low ER.

In terms of time, HL showed an increasing trend, and LH showed a decreasing trend in the first three periods. This result indicates that an increasing number of local governments preferred to increase EGC and relax ER during the 2000–2012 period. At the same time, HH and LL showed a trend toward fragmentation, suggesting that the mutual influence of the synergistic preference of local governments was weakening. In the last three periods, the number of HH increased again and showed agglomeration, and LL remained at a relatively high level, forming the weak synergistic preference of local governments. At the same time, HL showed a trend of decreasing and diversifying, and LH showed a trend of increasing, but there was a fluctuation in 2020. This result indicates that the behavioral preferences of some local governments began to shift toward reducing EGC and improving ER; however, the change was fluctuating.

In terms of space, in the first three periods, the clustering of HL and HH was the main feature in Anhui, and the clustering of LH and LL was the main feature in Jiangsu and Zhejiang. The behavioral preferences of local governments in the YRD region were generally characterized by clustering in patches. In the last three periods, the behavioral preferences of local governments in the YRD region were generally characterized by spatial diversification and small-scale clustering. Shanghai showed low EGC and high ER, except in 2000, when it showed low EGC and low ER.

### 4.2. PVAR

#### 4.2.1. Panel Unit Root Test and Determination of the Lag Order

The stationarity test is the premise of time series modeling. Nonstationary data will lead to spurious regression in the model estimation results, which will be unable to truly reflect the inherent logical relationship between variables. To avoid the appearance of spurious regression, referring to the previous literature [56], four unit root test methods LLC, IPS, Fisher ADF, and Fisher PP were used to test the stationarity of the data. After first-order difference processing of the data, the panel unit root test was performed to obtain the results (Table 4). All four tests rejected the hypothesis of the existence of the unit root. Thus, the data were stationary, and the PVAR model could be constructed.

The optimal lag order of the PVAR model should be determined before running the model. The AIC, BIC, and HQIC were selected to determine the optimal lag order which was chosen based on the minimum value [51]. The results show that the optimal lag orders determined by the AIC, BIC and HQIC are 2, 1, and 1, respectively (Table 5). Therefore, 1 was chosen as the optimal lag order.

#### 4.2.2. Impulse Response Analysis

The impulse response function simulates the dynamic influence trend of a unilateral positive standard deviation shock of one variable on another variable under the premise that other variables remain unchanged. In this study, an attempt was made to reflect the interaction mechanism among FP, EGC, ER and EGE through the mutual impact and response of variables. Impulse response results were obtained through Monte Carlo simulation involving 500 iterations (Figure 5).

Taking FP as a response object, the following occurs: (1) FP positively responds to its own shock in the lag 1 period and then begins to converge to 0, which shows that FP is self-reinforcing; (2) FP shows a negative response to the impact of EGC in the lag 1 period and then gradually converges to 0, which shows that EGC can alleviate FP; (3) after being impacted by ER, FP shows a negative response in the lag 1 period and then gradually converges to 0, indicating that ER can reduce FP; and (4) the impact of EGE on FP shows a negative response in the lag 1 period and then gradually converges to 0. That is, EGE can alleviate FP.

Taking EGC as a response object, the following occurs: (1) EGC negatively responds to its own shock in the lag 1 period and then gradually converges to 0, that is, EGC is self-weakening; (2) impacted by FP, EGC shows a negative response in the lag 1 period and then gradually converges to 0, indicating that FP can reduce EGC; (3) impacted by ER, EGC shows a negative response in the lag 1 period and a positive response in the lag 2 period, indicating that ER can reduce EGC and then increase EGC; and (4) EGC shows a positive response to the impact of EGE in the lag 1 period and then gradually converges to 0, indicating that EGE can enhance EGC.

Taking ER as a response object, the following occurs: (1) ER negatively responds to its own shock in the lag 1 period and then converges to 0, which shows that ER is self-weakening; (2) ER shows a negative response to the impact of FP in the lag 1 period and then converges to 0, indicating that FP can reduce ER; (3) impacted by ER, EGC shows a positive response in the lag 1 period and then gradually converges to 0, indicating that EGC can strengthen ER; and (4) impacted by EGE, ER in the lag 1 period shows a negative response, while in the lag 2 period, ER shows a positive response. This result indicates that the impact of EGE will reduce ER but that this reduction is not sustainable.

Taking EGE as a response object, the following occurs: (1) EGE negatively responds to its own shock in the lag 1 period and then gradually converges to 0, which shows that EGE is self-weakening; (2) the impact of FP on EGE shows a negative response in the lag 1 period and then gradually converges to 0, indicating that FP will reduce EGE; (3) impacted by EGC, EGE shows a positive response in the lag 0 period and converges to 0 in the lag 1 period, which shows that EGC can improve EGE and that this improvement occurs in the current period; and (4) EGE shows a positive response to the impact of ER in the lag 0 period and then gradually converges to 0, indicating that ER can also improve EGE and that this improvement occurs in the current period.

#### 4.2.3. Variance Decomposition Analysis

The purpose of variance decomposition is to examine the proportion of structural shocks to one variable from other variables. By analyzing the explanation and contribution of different variables, the degree of influence between variables can be described and measured more clearly. Since the relationship between the variables does not change significantly after the 6-period lag, 500 Monte Carlo simulations were used to obtain the variance decomposition results of the 1–6 lag periods (Table 6).

In the face of a structural shock, FP contributes on its own only at lag 1, and the rate of self-explanation remains at 83.3% at lag 6. This result shows that FP is characterized by self-dependence. As the lag increases, the degree of explanation of EGC, ER and EGE gradually increases. At lag 6, the explanation rates of ER and EGE are relatively strong, 6.65% and 9.88%, respectively. The explanation rate of EGC is relatively weak, 0.17%.

EGC also has a strong degree of self-explanation at lag 1 in the face of a structural shock. The rate of self-explanation is maintained at 80.51% at lag 6, indicating that EGC is also characterized by self-dependence. FP has an explanation rate of 8.68% at lag 1, while the explanation rate of the other variables is 0%, indicating that the impact of FP on EGC is more obvious and rapid. With the increase in the lag period, the explanation rate of FP continues to increase, reaching 15.29% at lag 6. The explanatory rates of ER and EGE are 2.63% and 1.57%, respectively. These results illustrate the relatively large degree to which FP affects EGC.

ER itself has a strong degree of self-explanation in the face of a structural shock at lag 1, and it remains at 93.03% at lag 6, indicating that ER has a strong degree of self-dependence. At lag 6, the explanation rate of EGC is 5.82%, while the explanation rates of FP and EGE are relatively weak, 0.58% and 0.57%, respectively. These results show that when facing a structural shock, ER is most influenced by itself, followed by EGC. The impacts of FP and EGE are relatively weak.

In the face of a structural shock, the degree of self-explanation of EGE gradually decreases with the increase in the lag period. The rate of self-explanation of EGE drops to 67.5% at lag 6, which shows that the self-dependence of EGE is weak. However, the degree of explanation of the other variables gradually increases with the increase in the lag period. At lag 6, ER has the strongest explanation rate, 29.46%, while FP and EGC have relatively weak explanation rates, 1.78% and 1.26%, respectively.

## 5. Discussion

### 5.1. Changes in FP and EGE

The results show that from 2000 to 2020, local governments in the YRD region were generally under FP with an increasing trend, which is consistent with previous studies [27,29,36]. A series of fiscal and taxation reforms, such as the tax-sharing system, income tax sharing, and the replacement of business tax with value-added tax, promoted by the Chinese central government, have reduced the fiscal revenue available to local governments [27]. In addition, in recent years, tax and fee reduction policies and the impact of COVID-19 have further reduced local government fiscal revenue [29]. Local finance relies heavily on central transfer payments, tax rebates and extra-budgetary revenue. However, Chinese local governments, which are characterized by quasi-totalism, have undertaken a considerable amount of public affairs, resulting in a large amount of fiscal expenditure. The imbalance of revenue and expenditure is the key reason for the FP of local governments.

In addition, we found that between 2000 and 2020, EGE in the YRD region showed a fluctuating and rising change trend. The fluctuating characteristics before 2012 were more obvious, and the rising characteristics after 2012 were more obvious. This result is similar to the findings of Hou et al. [57]. Since the 18th National Congress, China’s central government has vigorously promoted the national strategy of ecological civilization construction, introduced, and improved a large number of environmental policies and laws, and established supporting systems, such as the one-vote veto system and central environmental protection supervision. This makes environmental protection and governance an important constraint in the administrative assessment of local governments and the political promotion of local officials. We speculate that this result might be due to the promotion of the ecological civilization strategy and the high-level driving force of the central government. Moreover, a study found that the emissions of environmental pollutants in Zhejiang and Jiangsu showed a decreasing trend, while in Anhui Province, they increased [58]. However, we found that the average EGE in Anhui was higher than that in Zhejiang and Jiangsu. As a result, we believe that it is equally important to focus on the EGE rather than relying solely on changes in environmental pollution indicators to reflect the effectiveness of the environmental governance of local governments.

### 5.2. Changes in Local Government Behavioral Preferences

The results show that the behavioral preferences of local governments in the YRD changed with changes in institutions and policies. From a temporal perspective, during the 2000–2012 period, local governments in the YRD region were generally characterized by a strong alternative preference with the expansion of high EGC and low ER. This result is in line with the assumptions of the political promotion tournament and the pollution haven hypothesis. However, this characteristic changed after 2012. The impact of environmental authoritarianism reduced the applicability of the pollution haven hypothesis in the YRD region, while the impact of the political promotion tournament persisted. As a result, local government behavioral preferences showed the characteristic of weak synergy. We also found that the changes fluctuated.

The reasons for the weak synergistic preference are as follows: (1) The number of local governments with high EGC and high ER increased and clustered again, which means that some local governments were still facing greater EGC while improving ER. This situation indicates the continued impact of political promotion and administrative appraisal oriented toward economic growth [59]. (2) Local governments with low EGC and low ER still existed after 2012, indicating that there were still some local governments that did not fully implement the central government’s policy measures. At the same time, such strategic policy execution was often accompanied by low EGC. We speculate that this situation might be due to economic growth having a great and continuous incentive effect. Differences in the industrial structure and the existence of government-enterprise alliances resulted in fragmented interest games between the central and local governments [12,60], causing the high-level promotion of ER to not achieve good results in some areas. Low EGC also reflects that local governments achieved economic growth through this means.

The reasons for the fluctuation are that the number of local governments with high EGC and low ER decreased and was scattered, and the number of local governments with low EGC and high ER increased. The shock of changes in the alternative preference leads to fluctuations in the changes in the synergistic preference. We believe that the reason for this change in the alternative preference is that top-down decision-making and the resulting changes in institutions, structures, and policies can have a significant impact on the behavioral preferences of local governments in authoritarian China [40]. As a result, an increasing number of local governments in the YRD region began to strengthen ER driven by ecological civilization construction and the resulting array of policy tools and control measures [61]. At the same time, the Porter hypothesis may have applicability in the YRD region [62]. Therefore, the innovation compensation effect of ER promoted an increase in local GDP, which relieved the pressure of competition for economic growth. These factors created an increase in the number of local governments with low EGC and high ER. However, we found that the alternative preference fluctuated in 2020. This result may be due to the spillover effect of the innovation compensation effect of ER [63,64], which brings enterprise technological innovation and GDP growth in surrounding cities, leading to a renewed increase in EGC.

From a spatial perspective, the behavioral preferences of local governments in the YRD region between EGC and ER had spatial agglomeration characteristics. This finding indicates that there may be an imitation effect among local governments, which is consistent with the findings of Zhang et al. [65] and Pan et al. [66]. In addition, we found that the behavioral preferences of local governments showed the characteristics of large-scale agglomeration in the first three periods and diversification and small-scale agglomeration in the last three periods. We hold the view that environmental authoritarianism broke the regional solidification of local government behavioral preferences and formed a diversified spatial pattern in the YRD region. Meanwhile, local government behavioral preferences still exhibited agglomeration on a small scale due to the persistence of the imitation effect among local governments.

### 5.3. The Interaction Mechanism of FP, EGC, ER and EGE

Regarding the self-influence mechanism, the results show that FP, EGC, and ER have a strong self-dependence, indicating that there are path dependencies in changes in the YRD region. First, FP has a self-reinforcing effect, indicating that the FP faced by local governments in the YRD has been increasing, which is consistent with our findings above. Second, EGC has a self-weakening effect. The weakening of EGC means that the GDP growth rate of similar cities slowed down or that a city’s own GDP growth rate increased, indicating that the EGC of local governments in the YRD region is characterized by catching up and pressure in the political promotion tournament [67], which is similar to the view of Jiang et al. [68]. Third, the weakening effect of ER illustrates the dilemma faced by local governments in the YRD region in regard to maintaining a high level of ER, which is similar to the findings of Kou et al. [36]. The persistence of local governments with low ER in the findings above also reflects this reality. Fourth, EGE has not only a self-weakening effect but also a weak self-dependence. This result means that it is difficult for local governments to maintain a high level of EGE if they are not stimulated and driven by other factors, which reflects the limited capacity and lack of motivation of governments to govern the environment [69].

Regarding the interaction mechanism between FP and local government behavioral preferences in the YRD region, we found that (1) ER can reduce EGC, which in turn increases EGC. This result reveals that ER can increase the growth rate of local GDP, further validating the possibility that the Porter hypothesis exists in the YRD region. It also suggests that the innovation compensation effect of ER may have a spillover effect that stimulates a re-enhancement of EGC. In addition, the occurrence of these results has a lag effect over time. (2) EGC can enhance ER. As a result, we speculate that when an innovation compensation effect occurs, ER and economic growth will gradually achieve a win–win situation of development due to the persistence of the political promotion tournament [70]. (3) FP can reduce EGC, and the impact is more pronounced and rapid. This result indicates that FP leads local governments to take action to rapidly reduce the pressure of competition, which means that FP is a strong driving force for local governments to vigorously develop the economy. At the same time, the result also suggests that EGC can help to relieve FP. (4) The implementation of ER will not cause FP. Moreover, ER can help reduce FP significantly. We suspect that this result may be due to two reasons. First, limiting the emissions of enterprises can reduce the financial expenditure on pollution control. Second, ER can stimulate the technological innovation of enterprises, bringing profits and benefits that expand the tax base and tax sources of local governments. (5) FP can reduce ER. We found that in the YRD region, local governments affected by FP prefer to relax ER, but ER is in fact beneficial for alleviating FP and for economic growth. The key is whether ER can generate an innovation compensation effect. We suspect that the innovation compensation effect of ER may have a time lag and spatial spillover, which in turn generates the behavioral preferences of local governments. This result also points to the importance of promoting industrial structure upgrading and enterprise technological innovation for environmental protection and governance. (6) In addition, we found that the EGC among local governments in the YRD region does not lead to a reduction in ER, rather, is conducive to improving ER. FP is the main reason for local governments to relax ER.

In conclusion, there is a double negative interaction mechanism between FP and local government behavioral preferences in the YRD region from 2000 to 2020. That is, FP had a negative effect on EGC and ER, EGC and ER also had a negative effect on FP. But it is worth noting that the negative effect on FP is essentially a positive effect on the fiscal position of local governments. This means that FP made local governments prefer to reduce EGC and relax ER, but in fact, EGC and ER were conducive to alleviating FP. There are several studies [20,24,27,36] that point to the preferences for local government behavior under fiscal decentralization or FP, which are consistent with our findings. However, these studies did not analyze the inverse effect of EGC and ER on FP, while our study confirmed this inverse effect and found that local governments were in a state of misalignment between the subjective behavioral preferences and the perceptions of objective laws.

Regarding the interaction mechanism between FP, local government behavioral preferences and EGE, we found that (1) both EGC and ER can improve EGE and can be effective in the current period. The achievement of governance goals in China lies in setting correct incentive goals and pressure mechanisms for local governments through formal and informal institutions [70,71,72]. EGC is government behavior under the goal of political promotion incentives [71]. The pressure of punishment under the administrative assessment system has resulted in an enhancement of ER [72]. Our research results confirm that the incentives and pressures set by the central and upper-level governments are beneficial for improving EGE. However, we also found that the behavioral preferences of local governments in the YRD region showed the characteristics of increasingly diversified aggregation. As a result, we hold the view that achieving governance goals requires more attention and the strengthening of the correction and control of local governments’ deviation behaviors. (2) The impact of ER on EGE is more significant. Therefore, it can be considered that the high-level driving force of the central government is conducive to improving the efficiency of the environmental governance of local governments. This also explains why the EGE of local governments in the YRD region showed an upward trend after 2012. (3) FP can reduce EGC, ER, and EGE, while EGC and ER can improve EGE. We believe that the FP caused by the mismatch between financial power and administrative power makes the incentive and pressure measures of the central and upper-level governments have a deviation in the effect at the local level, resulting in a reduction of EGE. FP aggravates the behavioral preference of local governments to relax ER for the sake of economic growth. Therefore, the insufficient resources available for local governments to invest cannot lead to real governance effects. In this case, to improve the efficiency of local governments’ environmental governance, more attention should be paid to the solution to FP.

In conclusion, there is a negative transitive influence mechanism between FP, local government behavioral preferences and EGE in the YRD region from 2000 to 2020. That is, FP could have a negative effect on EGE and a negative effect on EGC and ER, but EGC and ER could have a positive effect on EGE. It can be deduced that the negative effect of FP on EGE can be transmitted by reducing EGC and ER. This means that FP can influence the EGE of local governments by changing their behavioral preferences between EGC and ER. There are studies [73,74] that found the adverse effects of FP on EGE, which are consistent with our findings. However, these studies paid less attention to whether the behavioral preferences of local governments intervene in this influence relationship. Based on the theory of the political promotion tournament and the pollution haven hypothesis, we defined the EGC and ER behaviors of local governments. We found that the change of local government behavioral preferences can play a transmission role in the influence of FP on EGE, which further expands the discussion of the influence mechanism between FP and EGE.

### 5.4. Policy Implications

Based on the discussion above, we note the following policy implications:(1)Pay attention to solving the problem of FP faced by local governments. We found that FP is the main reason for local governments to relax ER and that it is not conducive to improving the EGE. Therefore, attention should be paid to solving the problem of FP in the future. On the one hand, the central government should further improve the reform of the fiscal system, establish a fiscal system that matches its powers and expenditure responsibilities, and set up a fiscal security system to reduce the impact of systemic FP on local governments. At the same time, transfer payments should be increased, and the proportion of the main tax types between the central and local governments should be appropriately adjusted to increase the fiscal revenue of local governments. On the other hand, central and higher-level governments should scientifically divide their powers and regulate the functions and responsibilities of governments at all levels. Local governments should streamline administration, actively delegate power to lower levels, and deepen the reforms to improve government services. At the same time, social self-governing organizations should be encouraged and cultivated to reduce the fiscal expenditures of local governments.(2)Strengthen the control and correction of local government behavioral preferences. We found that the behavioral preferences of local governments in the YRD region showed a diversified trend and that there may be an interest game between the central and local governments. As a result, we suggest that the control and correction of local government behavioral preferences should be strengthened. First, local governments should be guided to set reasonable economic growth goals based on their own resource endowments and development advantages, and comprehensive competition from GDP to innovation, technology, industry, and other factors should be promoted to form high-quality economic growth. Second, it is necessary to strengthen the supervision of the implementation of strategic policies and improve and normalize the accountability and punishment systems, such as central inspections of environmental protection, to increase the cost of environmental violations by local governments. Finally, environmental adjustments to the performance assessment objectives should be made, and the quantitative design of the environmental assessment indicators should be optimized. It is necessary to establish and improve the performance assessment and incentive mechanism with hard environmental constraints as the core to enhance the motivation of local governments to engage in environmental protection and governance.(3)Promote the upgrading of the industrial structure and technological innovation of enterprises. We found that the innovation compensation effect of ER is the key to whether economic growth and ER can achieve a win–win situation. As a result, corresponding ERs and policies should be formulated for different regions and industries, especially those with high energy consumption, emissions, and pollution, which should be supervised and punished to force enterprises to conduct green technological innovation. At the same time, in the process of implementing ER, the government should provide policy support in terms of taxation and financing, encourage enterprises to increase their technological innovation to cope with the pressure of ER, strengthen investment in scientific research and guide enterprises to make technological changes to their production methods to enhance the technological content and added value of their products. Local governments and enterprises should be guided to transform from “competing for growth” to “competing for innovation”.(4)Improve the third-party governance of the environment. We found that local governments undertake a considerable number of public affairs and generally face FP; thus, it is difficult to maintain high EGE. Coupled with the government administrators’ lack of professional and technical knowledge with regard to environmental governance [69], local governments appear to be “powerless” in environmental governance. However, market-oriented third-party governance can effectively solve the problems of insufficient government pollution control capacity and the evasion of responsibility by polluting enterprises through specialization, scale, and technology [75]. Although the central government has issued policy documents to promote the third-party governance of environmental pollution, such a governance model is still in the initial stage of development based on the actual situation [76]. Therefore, further actions should be taken to promote and improve the third-party governance of environmental pollution.

## 6. Conclusions and Prospects

This study reveals the change characteristics of FP, local government behavioral preferences between EGC and ER, and EGE in the YRD region and analyzes the interaction mechanism between them. The conclusions are as follows:

(1) From 2000 to 2020, FP was generally strengthened. EGE generally showed fluctuating and rising change characteristics, with more obvious fluctuating and rising characteristics before 2012 and after 2012, respectively. Local governments shifted from a strong alternative preference to a weak synergistic preference. (2) FP had a self-reinforcing effect. EGC and ER had a self-weakening effect. EGE had not only a self-weakening effect but also a weak self-dependence. (3) There is a double negative interaction mechanism between FP and local government behavioral preferences. FP made local governments prefer to reduce EGC and relax ER, but in fact, EGC and ER were conducive to alleviating FP. Local governments were in a state of misalignment between the subjective behavioral preferences and the perceptions of objective laws. (4) There is a negative transitive influence mechanism between FP, local government behavioral preferences, and EGE. FP could have a negative effect on EGE, EGC, and ER, but EGC and ER could have a positive effect on EGE. The negative effect of FP on EGE can be transmitted by reducing EGC and ER.

This study provides a useful reference for improving the EGE of local governments in the YRD region and understanding the behavioral logic of local governments’ environmental governance. It can also provide a reference for other rapidly industrializing and urbanizing regions. However, there are also some limitations: (1) We proposed and discussed the changes in FP, local government behavioral preferences and EGE in the wake of institutional and policy changes. However, to ensure that the model analysis has a sufficient sample size to scientifically reflect the regularity of the interaction mechanism, we did not conduct a comparative analysis of the interaction mechanism before and after the institutional and policy changes, which will be the focus of our future research. (2) This study uses only the YRD region in China as an example and lacks evidence from other regions. Therefore, it is necessary to conduct empirical analysis on more regions in the future. (3) Since local governments are the main actors in environmental governance, this study focused on governmental behaviors in environmental governance. In the future, it is necessary to explore the interactions between the behaviors and decisions using an inter-disciplinary approach for the multiple actors (e.g., industry, consumers, NGOs) in environmental governance.

## Figures and Tables

**Figure 1 ijerph-19-16618-f001:**
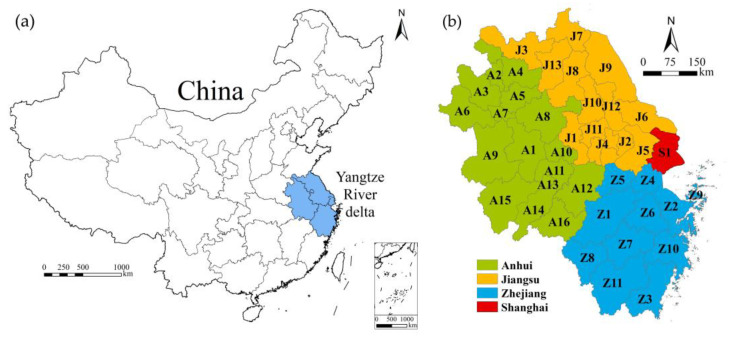
(**a**) Location of the YRD region in China. (**b**) Location of different provinces and cities in the YRD region. Note: This map is based on the standard map (Drawing review No.: GS (2019)1822).

**Figure 2 ijerph-19-16618-f002:**
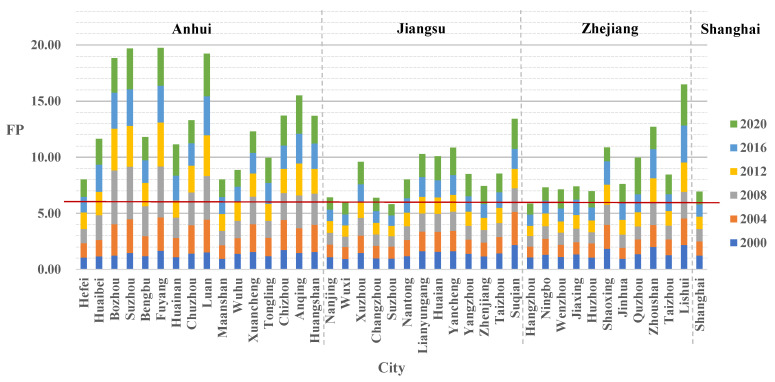
Changes in FP in the YRD region.

**Figure 3 ijerph-19-16618-f003:**
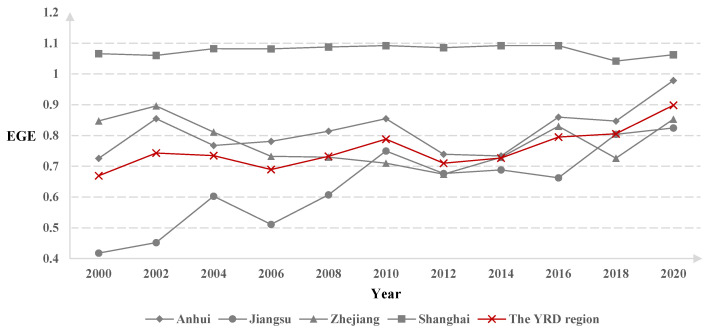
Changes in EGE in the YRD region.

**Figure 4 ijerph-19-16618-f004:**
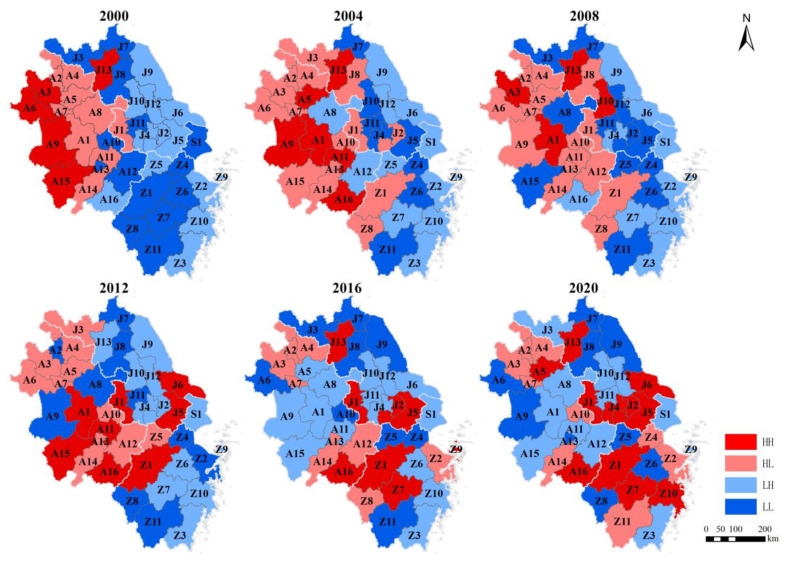
The spatiotemporal distribution of local government behavioral preferences.

**Figure 5 ijerph-19-16618-f005:**
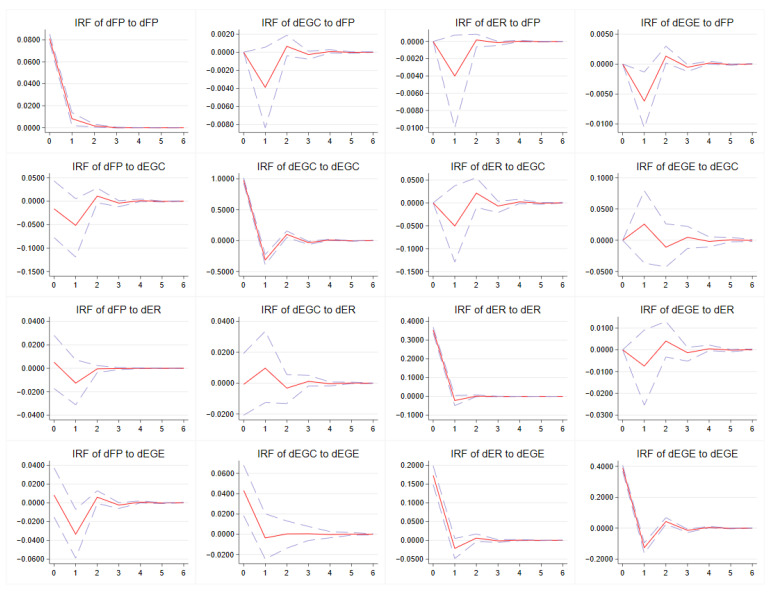
Impulse response results. Note: the horizontal axis is the lag order, and the vertical axis is the intensity of the response. The solid red line is the impulse response curve, and the dashed line indicates the upper and lower limits of the confidence interval.

**Table 1 ijerph-19-16618-t001:** City codes in the YRD region.

Anhui		Jiangsu		Zhejiang		Shanghai	
Hefei	A1	Nanjing	J1	Hangzhou	Z1	Shanghai	S1
Huaibei	A2	Wuxi	J2	Ningbo	Z2		
Bozhou	A3	Xuzhou	J3	Wenzhou	Z3		
Suzhou	A4	Changzhou	J4	Jiaxing	Z4		
Bengbu	A5	Suzhou	J5	Huzhou	Z5		
Fuyang	A6	Nantong	J6	Shaoxing	Z6		
Huainan	A7	Lianyungang	J7	Jinhua	Z7		
Chuzhou	A8	Huaian	J8	Quzhou	Z8		
Luan	A9	Yancheng	J9	Zhoushan	Z9		
Maanshan	A10	Yangzhou	J10	Taizhou	Z10		
Wuhu	A11	Zhenjiang	J11	Lishui	Z11		
Xuancheng	A12	Taizhou	J12				
Tongling	A13	Suqian	J13				
Chizhou	A14						
Anqing	A15						
Huangshan	A16						

Note: The order of city codes refers to the order in official and statistical yearbooks. The same applies below.

**Table 2 ijerph-19-16618-t002:** Descriptive statistics of the variables.

Variables	Number	Average	Standard Deviation	Max	Min
FP	861	1.7715	0.7837	5.1527	0.8520
EGC	861	98.4894	581.5932	10,335.0900	0.9200
ER	861	1.0035	0.6255	5.1700	0.0600
EGE	861	0.7374	0.3653	2.0527	0.1604

**Table 3 ijerph-19-16618-t003:** Moran’s I indices of EGC and ER in the YRD region.

Year	2000	2004	2008	2012	2016	2020
Moran’s I index	−0.0585	−0.0670	−0.1034	0.0104	−0.0359	0.0165
Spatial correlation	N	N	N	P	N	P

Note: N is negative, and P is positive.

**Table 4 ijerph-19-16618-t004:** Results of the panel unit root test.

Testing Method	dFP	dEGC	dER	dEGE
LLC	−3.9858 *** (0.0000)	−8.9846 *** (0.0000)	−4.8977 *** (0.0000)	−13.2104 *** (0.0000)
IPS	−1.7821 *** (0.0004)	−7.2924 *** (0.0000)	−3.6115 *** (0.0002)	−10.8355 *** (0.0000)
Fisher ADF	134.520 *** (0.0002)	238.237 *** (0.0000)	149.165 *** (0.0000)	257.500 *** (0.0000)
Fisher PP	134.082 *** (0.0003)	328.049 *** (0.0000)	141.045 *** (0.0001)	293.111 *** (0.0000)

Note: dFP, dEGE, dER, and dEGE represent the first-order difference after the variable takes the logarithm. *** indicates significance at the 1% level. The *p* values are in brackets.

**Table 5 ijerph-19-16618-t005:** Results of the multicriteria test.

Lag	AIC	BIC	HQIC
1	2.3302	3.4065	2.7442
2	2.3095	3.5322	2.7810
3	2.6075	3.9904	3.1422
4	3.6904	5.2496	4.2950
5	8.7170	10.4713	9.3992

**Table 6 ijerph-19-16618-t006:** Results of variance decomposition.

Response Variables	Lag Periods	Impulse Variables			
		dFP	dEGC	dER	dEGE
dFP	1	1.0000	0.0000	0.0000	0.0000
	2	0.9279	0.0017	0.0128	0.0576
	3	0.8868	0.0015	0.0301	0.0816
	4	0.8605	0.0014	0.0460	0.0921
	5	0.8336	0.0016	0.0668	0.0980
	6	0.8330	0.0017	0.0665	0.0988
dEGC	1	0.0868	0.9132	0.0000	0.0000
	2	0.1308	0.8355	0.0214	0.0123
	3	0.1425	0.8185	0.0260	0.0130
	4	0.1484	0.8109	0.0264	0.0143
	5	0.1514	0.8071	0.0263	0.0152
	6	0.1529	0.8051	0.0263	0.0157
dER	1	0.0076	0.0498	0.9426	0.0000
	2	0.0055	0.0552	0.9352	0.0041
	3	0.0051	0.0570	0.9325	0.0054
	4	0.0052	0.0578	0.9313	0.0057
	5	0.0055	0.0581	0.9307	0.0057
	6	0.0058	0.0582	0.9303	0.0057
dEGE	1	0.0165	0.0032	0.1755	0.8048
	2	0.0171	0.0089	0.2430	0.7310
	3	0.0177	0.0110	0.2730	0.6983
	4	0.0178	0.0120	0.2862	0.6840
	5	0.0178	0.0124	0.2920	0.6778
	6	0.0178	0.0126	0.2946	0.6750

## Data Availability

All data generated or analyzed during this study are included in the published article.

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
