# Peer review of "The Interaction Mechanism of Fiscal Pressure, Local Government Behavioral Preferences and Environmental Governance Efficiency: Evidence from the Yangtze River Delta Region of China"

_ijerph, 2022, doi:10.3390/ijerph192416618_

Round 1

Reviewer 1 Report

1.      The paper analyzes the interaction mechanism of fiscal pressure, government decisions and environmental governance efficiency between 2000 and 2020 in the Yangtze River Delta Region of China. Is well organized and provides numerous quantitative estimations.

2.      However, there are strong limitations regarding the scientific approach, originality, interest to readers (irrelevant results), and limited, untested conclusions.

3.      The research of the interaction mechanism is vague, confuse. Authors do not propose a conceptual model based on their results; although they claim so in the Conclusion section.

4.      Most indicators (e.g. intensity of environmental regulation) lack of thresholds or reference values that allow readers understand whether the condition is appropriate or not; if it is acceptable or unacceptable. Similarly, there is no precise quantification regarding how efficient the environmental governance is. Therefore, most equations are not realistic. A key constraint of the paper.

5.      The article does not follow environmental principles neither public health ones: both central issues for IJERPH.

6.      In 2022, environmental studies must follow an inter-disciplinary approach combining tools from natural sciences, social sciences and engineering. A modeling-statistical work is not good enough to investigate environmental governance. The research design of this paper lacks social studies: a strong bias. Besides, authors do not consider key social actors (e.g. industries, consumers, NGOs) other than governments. Social scientists should participate in this study to conduct ethnographic analysis by means of interviews, focal groups, and/or opinion surveys.

7.      Results do not necessary reflects the interaction among political decisions, industries decisions, individual decisions, public opinions, and the conditions of natural systems and people´s health.

8.      Authors arrive to some interesting conclusions but unsupported by sound data and results. “Business as usual” also occurs in China, although authors do not use this concept.

9.      Something relevant happened in 2012, but is not discussed in the paper.

10.  In summary, the article does not provide sound scientific knowledge. It requires a complete re-evaluation to improve results and add value to the work.

Author Response

We are very grateful to you for reviewing our manuscript and providing detailed comments. Your comments and suggestions are very helpful for revising and improving our paper, as well as the important guiding significance to our research. We have studied comments carefully and have made correction which we hope meet with approval. “Track Changes” function was used and revised portion were marked in red in the paper. The page numbers in parentheses below correspond to the version of manuscript with track changes. The main corrections in the paper and the responses to the reviewer’s comments are as flowing:

Reviewer 1:

The paper analyzes the interaction mechanism of fiscal pressure, government decisions and environmental governance efficiency between 2000 and 2020 in the Yangtze River Delta Region of China. Is well organized and provides numerous quantitative estimations. However, there are strong limitations regarding the scientific approach, originality, interest to readers (irrelevant results), and limited, untested conclusions.

In summary, the article does not provide sound scientific knowledge. It requires a complete re-evaluation to improve results and add value to the work.

Comment 1:

The research of the interaction mechanism is vague, confuse. Authors do not propose a conceptual model based on their results; although they claim so in the Conclusion section.

Response to comment 1:

Thank you very much for your comment. We apologize for not making the interaction mechanism clear. We fully agree that this suggestion can be of great worth to improve the value of our research. We followed your comments and made modifications (Page 17-18), mainly by conceptualizing and explaining the interaction mechanisms, elaborating their realistic manifestations, adding literature supports and comparison with the findings of others. The specific modifications are as follows:

Page 17

In conclusion, there is a double negative interaction mechanism between FP and local government behavioral preferences in the YRD region from 2000 to 2020. That is, FP had a negative effect on EGC and ER, EGC and ER also had a negative effect on FP. But it is worth noting that the negative effect on FP is essentially a positive effect on the fiscal position of local governments. This means that FP made local governments prefer to reduce EGC and relax ER, but in fact, EGC and ER were conducive to alleviating FP. There are several studies[20,24,27,36] that point to the preferences for local government behavior under fiscal decentralization or FP, which are consistent with our findings. However, these studies did not analyze the inverse effect of EGC and ER on FP, while our study confirmed this inverse effect and found that local governments were in a state of misalignment between the subjective behavioral preferences and the perceptions of objective laws.

References:

  1. Cai, H.; Tong, Z.; Xu, S.; Chen, S.; Zhu, P.; Liu, W. Fiscal Decentralization, Government Behavior, and Environmental Pollution: Evidence From China. Front. Environ. Sci. 2022, 10, 1–12, doi:10.3389/fenvs.2022.901079.
  2. Xiao-Sheng, L.; Yu-Ling, L.; Rafique, M.Z.; Asl, M.G. The Effect of Fiscal Decentralization, Environmental Regulation, and Economic Development on Haze Pollution: Empirical Evidence for 270 Chinese Cities during 2007–2016. Environ. Sci. Pollut. Res. 2022, 29, 20318–20332, doi:10.1007/s11356-021-17175-1.
  3. Bai, J.; Lu, J.; Li, S. Fiscal Pressure, Tax Competition and Environmental Pollution. Environ. Resour. Econ. 2019, 73, 431–447, doi:10.1007/s10640-018-0269-1.
  4. Kou, P.; Han, Y. Vertical Environmental Protection Pressure, Fiscal Pressure, and Local Environmental Regulations: Evidence from China’s Industrial Sulfur Dioxide Treatment. Environ. Sci. Pollut. Res. 2021, 28, 60095–60110, doi:10.1007/s11356-021-14947-7.

Page 18

In conclusion, there is a negative transitive influence mechanism between FP, local government behavioral preferences and EGE in the YRD region from 2000 to 2020. That is, FP could have a negative effect on EGE and a negative effect on EGC and ER, but EGC and ER could have a positive effect on EGE. It can be deduced that the negative effect of FP on EGE can be transmitted by reducing EGC and ER. This means that FP can influence the EGE of local governments by changing their behavioral preferences between EGC and ER. There are studies[73,74] that found the adverse effects of FP on EGE, which are consistent with our findings. However, these studies paid less attention to whether the behavioral preferences of local governments intervene in this influence relationship. Based on the theory of the political promotion tournament and the pollution haven hypothesis, we defined the EGC and ER behaviors of local governments. We found that the change of local government behavioral preferences can play a transmission role in the influence of FP on EGE, which further expands the discussion of the influence mechanism between FP and EGE.

References:

  1. Lin, B.; Zhou, Y. Does Fiscal Decentralization Improve Energy and Environmental Performance? New Perspective on Vertical Fiscal Imbalance. Appl. Energy 2021, 302, 117495, doi:10.1016/j.apenergy.2021.117495.
  2. Sun, Y.; Zhu, D.; Zhang, Z.; Yan, N. Does Fiscal Stress Improve the Environmental Efficiency? Perspective Based on the Urban Horizontal Fiscal Imbalance. Int. J. Environ. Res. Public Health 2022, 19, 6268, doi:10.3390/ijerph19106268.

Comment 2:

Most indicators (e.g. intensity of environmental regulation) lack of thresholds or reference values that allow readers understand whether the condition is appropriate or not; if it is acceptable or unacceptable. Similarly, there is no precise quantification regarding how efficient the environmental governance is. Therefore, most equations are not realistic. A key constraint of the paper.

Response to comment 2:

Thank you very much for your comment. We are very sorry for the confusion and we will explain these issues.

(1) Regarding the lack of thresholds or reference values for most indicators (e.g., intensity of environmental regulation). We fully agree that it is a very meaningful topic to further explore the thresholds, which is important in recognizing the appropriateness and acceptability of variables. We have carefully examined and discussed this suggestion, and we feel that this may detract from our research topic. The focus of our study is to explore the law of the interactions among fiscal pressure, local government behavioral preferences, and environmental governance efficiency using the PVAR model. In addition, we preliminarily analyzed the temporal trends of fiscal pressure and environmental governance efficiency, and combined environmental regulation with economic growth competition to reflect the spatio-temporal changes of local governments' behavioral preferences between economic development and environmental protection. It can be said that the trend of variables is not a priority of our study. The presence or absence of the threshold is also not an indispensable factor for our study. More importantly, the analysis of thresholds is a systematic work that requires a matching set of research methods and structures, which deviates from our research topic. As a result,the existence of thresholds for these variables was not calculated using other models due to the research framework, topic,structure, and space constraints. Without doubt, we fully agree that your comments are very valuable. They point the way to our future research. We plan to develop a new research framework and adopt new research methods to explore in depth the question of the existence of thresholds for these variables, and present our new findings to the readers. Thank you again for your valuable comments.

(2) Regarding the quantitative standards for environmental governance efficiency. First of all, we apologize for the confusion caused to you. The efficiency of environmental governance as defined in this study is mainly based on the inputs and outputs of environmental governance conducted by local governments, which is well suited to the use of the Super-SBM Model [1,2]. There have been many studies [3,4] that have used this approach to measure environmental governance efficiency. It is their ideas that we have taken into account in our study. In addition, we did not list our computational procedure in the paper because we consider space constraints and the fact that the model has been used several times in efficiency measurements. In the following, we will list our MATLAB code for the calculation using the Super-SBM Model. We hope it will be approved by you.

clc

clear

X =[ ];%Input Indicator Data

Y = [];%Desired Output Indicator Data

Z = [];%Data on non-desired output indicators

[m,n]=size(X);% m is the number of input indicators and n is the sample size

s=size(Y,1);%s is the desired number of outputs

q=size(Z,1);%q is the number of non-desired outputs

D=1./(m*X');%Denominator of the objective function

E=1./((s+q)*Y');%Equation constrains the denominator of the desired output

F=1./((s+q)*Z');%Equation constrains the denominator of the non-desired output

LB=zeros(n+m+s+q+1,1);UB=[];

theta=zeros(n,1);

for i=1:n

f1=[zeros(1,n) -D(i,:) zeros(1,s+q) 1];

A1=[];b1=[];

Aeq1=[X eye(m) zeros(m,s+q) -X(:,i)

     Y zeros(s,m) -eye(s) zeros(s,q) -Y(:,i)

     Z zeros(q,m) zeros(q,s) eye(q) -Z(:,i)

     zeros(1,n+m) E(i,:) F(i,:) 1];

beq1=[zeros(m,1)

     zeros(s,1)

     zeros(q,1)

      1];

[w1(:,i) theta(i)]=linprog(f1,A1,b1,Aeq1,beq1,LB,UB);

if abs(theta(i)-1)<0.00001   

f=[zeros(1,n) D(i,:) zeros(1,s+q) 1];

A=[[X(:,1:i-1),zeros(m,1),X(:,i+1:n)] -eye(m) zeros(m,s+q) -X(:,i)

   - [Y(:,1:i-1),zeros(s,1),Y(:,i+1:n)] zeros(s,m) -eye(s) zeros(s,q) Y(:,i)

   [Z(:,1:i-1),zeros(q,1),Z(:,i+1:n)] zeros(q,m+s)  -eye(q) -Z(:,i)];

b=zeros(m+s+q,1);

Aeq=[zeros(1,n+m) -E(i,:) -F(i,:) 1];

beq=1;

[w(:,i) theta(i)]=linprog(f,A,b,Aeq,beq,LB,UB);

end

end

theta%Value of efficiency

References:

  1. Tone, K. A Slacks-Based Measure of Super-Efficiency in Data Envelopment Analysis. Eur. J. Oper. Res. 2002, 143, 32–41, doi:10.1016/S0377-2217(01)00324-1.
  2. Li, H.; Fang, K.; Yang, W.; Wang, D.; Hong, X. Regional Environmental Efficiency Evaluation in China: Analysis Based on the Super-SBM Model with Undesirable Outputs. Math. Comput. Model. 2013, 58, 1018–1031, doi:10.1016/j.mcm.2012.09.007.
  3. Zang, J.; Liu, L. Fiscal Decentralization, Government Environmental Preference, and Regional Environmental Governance Efficiency: Evidence from China. Ann. Reg. Sci. 2020, 65, 439–457, doi:10.1007/s00168-020-00989-1.
  4. Zhang, Y.; Shen, L.; Shuai, C.; Bian, J.; Zhu, M.; Tan, Y.; Ye, G. How Is the Environmental Efficiency in the Process of Dramatic Economic Development in the Chinese Cities? Ecol. Indic. 2019, 98, 349–362, doi:10.1016/j.ecolind.2018.11.006.

Comment 3:

The article does not follow environmental principles neither public health ones: both central issues for IJERPH.

Response to comment 3:

Thank you for this comment. Our paper attempts to open the black box of government behavior in environmental governance and provides a scientific basis for improving environmental governance efficiency in the YRD region, harmoniously developing the economy and environment, and promoting ecological civilization construction. We believe that our research is in line with the scope of the journal IJERPH.

Here are our reasons. Firstly, as a public good, the environment is nonexclusive, noncompetitive and external. The protection and governance are mainly provided by governments. Secondly, environmental problems are mainly caused by pollution emissions from industrial enterprises in China. However, industry supports local economic growth and GDP is an important indicator in the assessment system for local governments and officials in China. Moreover, the tax payments of industrial enterprises are an important source of local fiscal revenue. Therefore, local governments that lack restrictions and constraints tend to relax environmental regulation, thereby seeking economic growth and increased fiscal revenues. This preference of local governments is an important reason why environmental pollution is endless and difficult to completely eradicate. Thirdly, as mega public organizations, governments need to regain their focus on the value of efficiency in the modernization process. Measuring the environmental governance efficiency from the perspective of inputs and outputs is conducive to reflecting the effectiveness of environmental governance in a more scientific and comprehensive manner. Fourthly, this study measured local government FP, EGC, ER and EGE in China’s Yangtze River Delta region from 2000 to 2020. Moran’s I index was used to identify the change characteristics of local government behavioral preferences. The interaction mechanism was analyzed by a panel vector autoregression model. Based on the above, this article discussed the findings and made policy recommendations.

We submitted this paper to the special issue "Environmental Protection Behavior: Strategies for Formation and Recurrence", which focuses on topics such as "relationship between environmental protection behavior and current major social problems, impacts of environmental protection behavior, sustainable utilization of resources and environmental governance from a multidisciplinary perspective." These are consistent with our research issue.

Comment 4:

In 2022, environmental studies must follow an inter-disciplinary approach combining tools from natural sciences, social sciences and engineering. A modeling-statistical work is not good enough to investigate environmental governance. The research design of this paper lacks social studies: a strong bias. Besides, authors do not consider key social actors (e.g. industries, consumers, NGOs) other than governments. Social scientists should participate in this study to conduct ethnographic analysis by means of interviews, focal groups, and/or opinion surveys.

Comment 5:

Results do not necessary reflects the interaction among political decisions, industries decisions, individual decisions, public opinions, and the conditions of natural systems and people´s health.

Response to comment 4-5:

Thank you very much for your comment. We agree that an inter-disciplinary approach can contribute to a more comprehensive analysis of environmental governance. We have carefully examined and discussed this suggestion. The following is our response.

(1) About " A modeling-statistical work is not good enough to investigate environmental governance. The research design of this paper lacks social studies: a strong bias."

The generation of our research ideas and the selection of our research methods build on a number of existing studies. For example, Bai et al. argued that Chinese local governments have little or no formal taxation power. It is more appropriate to explain environmental pollution from the perspective of fiscal concentration and FP. Additionally, they analyzed the negative impact of local government tax competition on environmental quality using data from 30 provinces in China [1]. Kou et al. held a similar view and examined the local gov-ernment regulation of SO2 under dual environmental pressure and FP using data from 30 provinces in China [2]. In addition, Zhang et al. confirmed a positive relationship between the scale of local government debt and environmental pollution based on Chinese provincial panel data [3]. Chang et al. demonstrated that local government FP can increase air pollution using panel data from resource-dependent cities in China [4]. Kong et al. found that the financial status of local governments in China is an important driver of environmental pollution control [5].

There are similarities between the above studies and our research topic, framework, and content, and all of these studies used modeling statistics. Therefore, we chose the method of modeling statistics to conduct our study. In addition, we are concerned with the efficiency of environmental governance and the interaction mechanism, which are more suitable for measurement and analysis using the modeling statistics approach. Although the modeling statistics approach has some shortcomings, it has advantages in providing intuitive and reliable evidence. We recognize the importance and value of social research methods, and we have used numerous field studies and case studies in our other daily research work. For this paper, however, we developed a research framework, and the statistical modeling approach is more conducive to measuring these variables and providing evidence of the interaction mechanism.

(2) About " Authors do not consider key social actors (e.g. industries, consumers, NGOs) other than governments. Social scientists should participate in this study to conduct ethnographic analysis by means of interviews, focal groups, and/or opinion surveys."

The efficiency of local governments' environmental governance is the focus of our study. This means that within the framework of our study, it is mainly a set of actions of local governments that are analyzed. In the context of China's promotion of pluralistic governance actors, other key social actors are indeed what should be focused on in environmental governance research, but are not the contribution and significance of this paper. Unlike in Western countries, in authoritarian China, the emergence and development of environmental governance follows a top-down path, meaning that government actions have a very strong influence on environmental issues. Therefore, we analyze the behavioral preferences of local governments and the efficiency of local governments' environmental governance, which are the starting points of our study. It is also due to China's unique institutional logic and governance model that, when conducting research on some important and sensitive issues, we found that interviews or observations of government officials did not yield completely true, valid and sufficient information and evidence. This is one of the important reasons why we started to try to seek objective data and try to use modeling statistics to conduct empirical research. Of course, inter-disciplinary research is our common expectation and goal. Your comments gave us important inspiration and we will try to design a inter-disciplinary research approach in our next study to analyze multiple actors in environmental governance.

(3) About "Results do not necessary reflects the interaction among political decisions, industries decisions, individual decisions, public opinions, and the conditions of natural systems and people´s health."

As we explained above, the behavior of local governments is the focus of our study. Therefore, we did not focus on other actors in environmental governance in this study, and did not produce relevant results. We are totally agree with you and will continue to pay attention to what you mentioned in future studies. In addition, we added the relevant expressions of shortcomings and perspectives at the end of this article based on your opinions (Page 20).

“Since local governments are the main actors in environmental governance, this study focused on governmental behaviors in environmental governance. In the future, it is necessary to explore the interactions between the behaviors and decisions using an interdisciplinary approach for the multiple actors (e.g., industry, consumers, NGOs) in environmental governance.”

References:

1.Bai, J.; Lu, J.; Li, S. Fiscal Pressure, Tax Competition and Environmental Pollution. Environ. Resour. Econ. 2019, 73, 431–447. https://doi.org/10.1007/s10640-018-0269-1.

2.Kou, P.; Han, Y. Vertical Environmental Protection Pressure, Fiscal Pressure, and Local Environmental Regulations: Evidence from China’s Industrial Sulfur Dioxide Treatment. Environ. Sci. Pollut. Res. 2021, 28, 60095–60110. https://doi.org/10.1007/s11356-021-14947-7.

3.Zhang, Z.; Zhao, W. Research on Financial Pressure, Poverty Governance, and Environmental Pollution in China. Sustaina-bility 2018, 10, 1834. https://doi.org/10.3390/su10061834.

4.Hui, C.; Shen, F.; Tong, L.; Zhang, J.; Liu, B. Fiscal Pressure and Air Pollution in Resource-Dependent Cities: Evidence From China. Front. Environ. Sci. 2022, 10, 908490. https://doi.org/10.3389/fenvs.2022.908490.

5.Kong, D.; Zhu, L. Governments’ Fiscal Squeeze and Firms’ Pollution Emissions: Evidence from a Natural Experiment in China; Springer: Dordrecht, The Netherlands, 2022; Volume 81; ISBN 0123456789.

Comment 6:

Authors arrive to some interesting conclusions but unsupported by sound data and results. “Business as usual” also occurs in China, although authors do not use this concept.

Response to comment 6:

Thank you very much for your comment. The following is our response.

(1) About “Authors arrive to some interesting conclusions but unsupported by sound data and results.”

First of all, we apologize for the confusion we caused you. In this study, the 2000–2020 panel data covering 41 cities in the YRD region are mainly from the “Anhui Statistical Yearbook”, “Jiangsu Statistical Yearbook”, “Zhejiang Statistical Yearbook” and “Shanghai Statistical Yearbook”. They are supplemented by the “China Urban Statistical Yearbook”, “China Environmental Statistical Yearbook” and the statistical yearbooks of various cities. In addition,moran’s I index was used to identify the change characteristics of local government behavioral preferences. The interaction mechanism was analyzed by a panel vector autoregression model. The above data and methods are more frequently used in relevant empirical studies. In the Results, we analyzed the results based on data and models, including“Trends in FP and EGC”, “Characteristics of Local Government Behavioral Preferences”, “Panel Unit Root Test and Determination of the Lag Order”, “Impulse Response Analysis”, “Variance Decomposition Analysis”. On these bases, we obtained the results of our study.

(2) About “Business as usual” also occurs in China, although authors do not use this concept.”

In this study, we conducted a general pattern analysis for the entire Yangtze River Delta region. As you mentioned, although policy designation and institutional improvement by the central government have increased significantly after the 18th National Congress, some local governments still have deviations in environmental policy implementation (Page15-16). For example, we found that local governments in some regions continue to face greater competitive pressures for economic growth while upgrading environmental regulations, with the continued influence of political promotion and administrative appraisal systems that target economic growth. At the same time, there were still some local governments that did not fully implement and enforce the central government's environmental policy measures. We suspected that this is due to the strong incentives of political promotion, the differences in industrial structure and the existence of political-enterprise alliances that created a fragmented game of interests between the central and local governments, resulting in which the environmental regulation did not work well in some regions, seemingly “business as usual”. However, the focus of this study is not to analyze these phenomena in depth. We plan to conduct a study to analyze why local governments exhibit both policy implementation innovation and policy implementation bias, taking into account the influence of other actors in policy implementation (e.g., industries, consumers, NGOs), and to search for the logic of local government behavior in environmental governance. We hope that the results of this future study can answer the question of the "business as usual" phenomenon when they are presented to the readers.

Comment 7:

Something relevant happened in 2012, but is not discussed in the paper.

Response to comment 7:

Thank you very much for your comment. As you suggested, we added the relevant description in the discussion section (Page 15). The specific modifications are as follows:

Since the 18th National Congress, China’s central government has vigorously promoted the national strategy of ecological civilization construction, introduced and improved a large number of environmental policies and laws, and established supporting systems, such as the one-vote veto system and central environmental protection supervision. This makes environmental protection and governance an important constraint in the administrative assessment of local governments and the political promotion of local officials.

We tried our best to improve the manuscript and made some changes in the manuscript. These changes will not influence the content and framework of the paper. We appreciate for your warm work earnestly and hope that the correction and respond will meet with approval. Once again, thank you very much for your comments and suggestions.

Best regards,

Tinghui Wang, Qi Fu, and Jinhua Chen

Reviewer 2 Report

This article is generally considered complete in terms of structure, research methods, etc. 

In this study on environmental governance, local governments are the main actors. 

Their behavioral preferences between economic growth competition (EGC) and environmental regulation (ER) affect the inputs and outputs of ecological governance. In general, the manuscript has been prepared very honestly. However, some issues need to be supplemented.

Your conclusions should be compared with those of others to get a better overall research value. It is suggested that in the conclusion part, some literature support is needed, which will improve the reference value of this article.

If the authors can improve the above suggestions, this article's reference value will significantly improve.

Author Response

We are very grateful to you for reviewing our manuscript and providing detailed comments. Your comments and suggestions are very helpful for revising and improving our paper, as well as the important guiding significance to our research. We have studied comments carefully and have made correction which we hope meet with approval. “Track Changes” function was used and revised portion were marked in red in the paper. The page numbers in parentheses below correspond to the version of manuscript with track changes. The main corrections in the paper and the responses to the reviewer’s comments are as flowing:

Reviewer 2:

This article is generally considered complete in terms of structure, research methods, etc. In this study on environmental governance, local governments are the main actors. Their behavioral preferences between economic growth competition (EGC) and environmental regulation (ER) affect the inputs and outputs of ecological governance. In general, the manuscript has been prepared very honestly. However, some issues need to be supplemented.

Comment 1:

Your conclusions should be compared with those of others to get a better overall research value. It is suggested that in the conclusion part, some literature support is needed, which will improve the reference value of this article. If the authors can improve the above suggestions, this article's reference value will significantly improve.

Response to comment 1:

We would like to express our sincere thanks to you for your suggestions. Your suggestions are significant for the improvement of the reference value of our article. We have revised it accordingly (Page 16-18). The specific modifications are as follows:

(1) For the self-influence mechanism of the variables, we added two literature supports (Page 16).

indicating that the EGC of local governments in the YRD region is characterized by catching up and pressure in the political promotion tournament [67], which is similar to the view of Jiang et al.[68]. Third, the weakening effect of ER illustrates the dilemma faced by local governments in the YRD region in regard to maintaining a high level of ER, which is similar to the findings of Kou et al.[36].

References:

  1. Kou, P.; Han, Y. Vertical Environmental Protection Pressure, Fiscal Pressure, and Local Environmental Regulations: Evidence from China’s Industrial Sulfur Dioxide Treatment. Environ. Sci. Pollut. Res. 2021, 28, 60095–60110, doi:10.1007/s11356-021-14947-7.
  2. Jiang, Y.P.; Waley, P.; Gonzalez, S. Shanghai Swings: The Hongqiao Project and Competitive Urbanism in the Yangtze River Delta. Environ. Plan. A-ECONOMY Sp. 2016, 48, 1928–1947, doi:10.1177/0308518X16652897.

(2) For the interaction mechanism between FP and local government behavioral preferences, we added four literature supports and compared with them (Page 17). The details are as follows:

There are several studies[20,24,27,36] that point to the preferences for local government behavior under fiscal decentralization or FP, which are consistent with our findings. However, these studies did not analyze the inverse effect of EGC and ER on FP, while our study confirmed this inverse effect and found that local governments were in a state of misalignment between the subjective behavioral preferences and the perceptions of objective laws.

References:

  1. Cai, H.; Tong, Z.; Xu, S.; Chen, S.; Zhu, P.; Liu, W. Fiscal Decentralization, Government Behavior, and Environmental Pollution: Evidence From China. Front. Environ. Sci. 2022, 10, 1–12, doi:10.3389/fenvs.2022.901079.
  2. Xiao-Sheng, L.; Yu-Ling, L.; Rafique, M.Z.; Asl, M.G. The Effect of Fiscal Decentralization, Environmental Regulation, and Economic Development on Haze Pollution: Empirical Evidence for 270 Chinese Cities during 2007–2016. Environ. Sci. Pollut. Res. 2022, 29, 20318–20332, doi:10.1007/s11356-021-17175-1.
  3. Bai, J.; Lu, J.; Li, S. Fiscal Pressure, Tax Competition and Environmental Pollution. Environ. Resour. Econ. 2019, 73, 431–447, doi:10.1007/s10640-018-0269-1.
  4. Kou, P.; Han, Y. Vertical Environmental Protection Pressure, Fiscal Pressure, and Local Environmental Regulations: Evidence from China’s Industrial Sulfur Dioxide Treatment. Environ. Sci. Pollut. Res. 2021, 28, 60095–60110, doi:10.1007/s11356-021-14947-7.

(3) For the interaction mechanism between FP, local government behavioral preferences and EGE, we added two literature supports and compared with them (Page 18). The details are as follows:

There are studies[73,74] that found the adverse effects of FP on EGE, which are consistent with our findings. However, these studies paid less attention to whether the behavioral preferences of local governments intervene in this influence relationship. Based on the theory of the political promotion tournament and the pollution haven hypothesis, we defined the EGC and ER behaviors of local governments. We found that the change of local government behavioral preferences can play a transmission role in the influence of FP on EGE, which further expands the discussion of the influence mechanism between FP and EGE.

References:

  1. Lin, B.; Zhou, Y. Does Fiscal Decentralization Improve Energy and Environmental Performance? New Perspective on Vertical Fiscal Imbalance. Appl. Energy 2021, 302, 117495, doi:10.1016/j.apenergy.2021.117495.
  2. Sun, Y.; Zhu, D.; Zhang, Z.; Yan, N. Does Fiscal Stress Improve the Environmental Efficiency? Perspective Based on the Urban Horizontal Fiscal Imbalance. Int. J. Environ. Res. Public Health 2022, 19, 6268, doi:10.3390/ijerph19106268.

We tried our best to improve the manuscript and made some changes in the manuscript. These changes will not influence the content and framework of the paper. We appreciate for your warm work earnestly and hope that the correction will meet with approval. Once again, thank you very much for your comments and suggestions.

Best regards,

Tinghui Wang, Qi Fu, and Jinhua Chen
